# Modeling of the Peptide Release during Proteolysis of β-Lactoglobulin by Trypsin with Consideration of Peptide Bond Demasking

**DOI:** 10.3390/ijms241511929

**Published:** 2023-07-25

**Authors:** Mikhail M. Vorob’ev

**Affiliations:** A.N. Nesmeyanov Institute of Organoelement Compounds, Russian Academy of Sciences, 28 ul. Vavilova, 119991 Moscow, Russia; mmvor@ineos.ac.ru

**Keywords:** proteolysis mechanisms, trypsin, peptide release, demasking kinetics

## Abstract

Prospects for predicting the fragmentation of polypeptide chains during their enzymatic hydrolysis using proteolysis models are considered. The opening of the protein substrate during proteolysis and the exposure of its internal peptide bonds for a successful enzymatic attack, the so-called demasking process, were taken into account. The two-step proteolysis model was used, including the parameters of demasking and the rate constants of hydrolysis of enzyme-specific peptide bonds. Herein, we have presented an algorithm for calculating the concentrations of intermediate and final peptide fragments depending on the time of hydrolysis or the degree of hydrolysis. The intermediate peptide fragments with two or one internal specific peptide bond were considered. The fragmentation of β-lactoglobulin (β-LG) by trypsin was predicted, and the calculated concentration curves for peptide fragments were compared with the experimental dependences of the concentrations on the degree of hydrolysis. Numerical parameters were proposed that characterize the concentration curves for intermediate and final peptide fragments, and they were used to compare the calculated and experimental dependences. The predicted distribution of the peptide fragments corresponded to the experimental data on the peptide release during the proteolysis of β-LG by trypsin.

## 1. Introduction

The enzymatic hydrolysis of proteins (proteolysis) leads to the formation of a mixture of various peptide fragments, the composition of which continuously changes during hydrolysis. The final products of proteolysis that do not include enzyme-specific peptide bonds, as well as sufficiently long intermediate peptides containing specific bonds, can be biologically active. To obtain the final products of proteolysis, it is necessary to carry out proteolysis to the end. The intermediate peptides can be obtained by proteolysis if the time interval is known, during which the reaction should be stopped in order to avoid further degradation. An aid in the production of bioactive peptides by proteolysis can be the prediction of peptide release using quantitative proteolysis models and computer simulations.

It is clear that to predict the release of the peptide during proteolysis, it is necessary to know the quantitative contribution to the specificity of the amino acid residues located in the corresponding cleaved sites. The combination of the contributions of amino acid residues at different positions in the hydrolyzable bond determines the probability of its cleavage within the framework of primary or secondary specificity [1,2]. In addition to knowing the specificity parameters, it is necessary to know whether the peptide chain conformation is convenient for a successful enzymatic attack of a given peptide bond, i.e., is this bond demasked or not [3]. When modeling such a complex phenomenon as proteolysis, various simplifications in the calculation algorithm are inevitable, including a decrease in the number of intermediate peptide fragments and an approximate determination of kinetic parameters that cannot be determined experimentally. Herein, we propose a new algorithm for predicting the release of peptides during β-LG proteolysis with trypsin, taking into account the process of peptide bond demasking.

The first quantitative data on the secondary specificity has been obtained for pepsin, considering up to 10 amino acid residues on both sides of the broken bond [4]. These data were used for the prediction of the peptide release in β-LG with demasked peptide bonds [5]. The efforts are currently ongoing to analyze the secondary specificity of pepsin and to model the peptide release kinetics [6,7]. For trypsin, mostly primary substrate specificity was taken into consideration for the proteolysis modeling [2,8], although the preferred and undesirable amino acid residues at various positions have also been established and considered [9]. Among the modeling of peptide release, we also note the studies of proteolysis with the proteases from *Bacillus licheniformis* [10] and *Lactococcus lactis* [11]. In all these works, it was assumed that any part of the polypeptide chain is freely accessible to the enzyme, or some constant factor was introduced to assess the limited accessibility of peptide bonds. No attempts have been made to evaluate the change in peptide bond accessibility during proteolysis, which means that these studies have not considered the process of demasking.

In the protein substrates, trypsin cleaves predominantly the peptide bonds at the carboxyl side of lysine and arginine (Arg-X and Lys-X bonds) unless they are followed by proline [12,13,14]. In addition, the rate of hydrolysis of these bonds also depends on other neighboring amino acid residues [9]. The hydrolysis of some peptide bonds during proteolysis is accompanied by structural changes in the protein, which, in turn, predetermine the hydrolysis of other bonds. Therefore, the opening of a protein substrate during proteolysis and the corresponding exposure of its internal peptide bonds for a successful enzymatic attack, the so-called demasking effect, is a very important component of proteolysis that controls its overall kinetics [15,16]. In contrast to the hydrolysis of low-molecular-weight substrates with a single hydrolyzable bond, proteolysis cleaves a set of peptide bonds with different secondary specificities and states of demasking, the latter being able to change during the process [15,16]. 

Proteolysis begins with the destruction of the original structure of a globular protein or protein aggregates (micelles), leading to an increase in the accessibility of peptide bonds for the enzyme. This process provides the demasking of peptide bonds, which leads to an increase in the total rate of hydrolysis when initially masked bonds become demasked [3]. The process opposite to demasking was observed during proteolysis of β-CN with trypsin, when some time after the onset of proteolysis, increased aggregation and a local enhancement of masking were observed [16]. The formation of additionally masked peptide bonds from proteolysis intermediates as a result of their aggregation or conformational rearrangements is referred to as “secondary masking”. It has recently been shown that the competition between peptide bond demasking and secondary masking explains the limitation of substrate structure opening and the restriction in peptide bond hydrolysis [17] with decreasing enzyme concentration.

The hydrolysis of peptide bonds was studied using the two-step proteolysis model that takes into account the demasking process [3,16]. The rate of demasking was determined by the shift in tryptophan fluorescence, which changes as the protein globule degrades or protein micelles are destroyed [15,16]. A complication of the two-step proteolysis model was carried out for proteolysis of β-LG by trypsin, considering two different stages of demasking, corresponding to the degradation of the protein globule and the destruction of the remaining hydrophobic core [18].

There are two possibilities for describing proteolysis: either in terms of peptide fragments or peptide bonds. Depending on the time of proteolysis, the concentrations of the peptide fragments or enzyme-specific peptide bonds can be analyzed. Since there are many more peptide fragments than specific peptide bonds, the description of proteolysis in terms of peptide fragments is more detailed. This description provides more data on the kinetics of proteolysis, although it is more laborious and some of the information may be redundant because it is not known how to interpret it. The concentrations of peptide bonds that have not been hydrolyzed at a given time of hydrolysis, depending on this time, contain much less information. Knowing the concentrations of all peptide fragments, it is possible to calculate the concentrations of peptide bonds, but an exact reverse calculation is impossible. It was shown that even with the loss of some peptide fragments, it is possible to reliably determine the concentration of peptide bonds according to HPLC-MS data [10,19]. It has also been shown that the changes in the concentrations of peptide bonds reflect the processes of demasking for these bonds [20].

Herein, we show how proteolysis can be described in terms of fragment concentrations with the consideration of the demasking of peptide bonds. The concentrations of peptides (final and intermediate products of proteolysis) were presented as functions of the degree of hydrolysis. The simulation results were compared with the experimental data [21,22,23,24,25] on the release of peptides during the hydrolysis of β-LG by trypsin. Our goal was to present in detail the procedure for choosing the calculation parameters, including the demasking and hydrolysis rate constants. The equations for calculating the concentration of peptides were also presented, and the parameters of the concentration curves were considered, which made it possible to compare the simulation results with the experiment.

## 2. Results

### 2.1. The Fragmentation with One Demasking Step

Proteolysis of a protein substrate is described by specifying the pathway for the cleavage of long fragments of the polypeptide chain into shorter ones. The concentrations of these fragments depending on the hydrolysis time are obtained by solving a system of differential equations describing the fragmentation kinetics, considering the material balance equations. In the case when all peptide fragments and all peptide bonds are freely available for the action of the enzyme and, therefore, there is no masking, the solution of the kinetic task is trivial [26]. If some of the fragments are masked and the peptide bonds in them are inaccessible to the enzyme, then the analysis becomes difficult. One possible way to simplify the task is presented in Figure 1, which schematically shows the formation and further hydrolysis of a trimeric block containing two enzyme-specific peptide bonds.

It is assumed that the ABC region of the polypeptide chain located between the most rapidly hydrolyzed peptide bonds opens up for enzymatic attack as a result of one-stage demasking [18] (Figure 1). The rate of hydrolysis of the internal peptide bonds in this region is controlled by demasking, i.e., the effective rate constants of their hydrolysis are equal to or less than the rate constant of demasking $k_{d}^{f}$. The size of the demasked region here is limited to three blocks of amino acid sequences, although kinetic patterns for a longer region can be derived in a similar way. According to our simplification, hydrolysis of the bonds with indexes *i* (A-B bond) and *j* (B-C bond) is impossible in the original polypeptide chain but is possible only in the ABC trimer and in the AB and BC dimers. Hydrolysis of the demasked peptide bonds A-B and B-C occurs with the hydrolysis rate constants *k^i^* and *k^j^*, resulting in the formation of the fragments AB, BC, A, B, and C.

In the scheme shown in Figure 1, the demasking process results in both the release of the ABC molecule and the opening of the A-B and B-C peptide bonds. The dependences of the concentrations on time for all fragments are given in Section 4.1 (Equations (1)–(6)). The initial concentrations of amino acid sequences in any position of the polypeptide chain are taken as a unit. For example, for the fragments containing the sequence A, the material balance equation [A] + [AB] + [ABC] + [-ABC-] = 1 is valid. Thus, the concentrations of all peptide fragments given here are relative.

### 2.2. The Fragmentation with Two Demasking Steps

It has been shown that some sites are demasked in two stages, since after the first stage of demasking with the rate constant of demasking $k_{d}^{f}$, they are still in a hydrolysis-resistant core [18]. The second demasking step with the rate constant *k_d_* yields the demasked trimer ABC containing two demasked enzyme-specific peptide bonds (Figure 2).

In this scheme (Figure 2), the two-stage demasking process results in both the release of ABC molecule and the opening of the ith and jth peptide bonds. The hydrolysis of the ABC trimer proceeds in the same way as in the previous scheme (Figure 1). The time dependences for the concentrations of all fragments are given in Section 4.2 (Equation (7)).

### 2.3. Application of Peptide Release Schemes to β-LG Proteolysis by Trypsin

To apply the model schemes (Figure 1 and Figure 2) to real proteolysis, we collected here two sets of kinetic parameters for the enzyme-specific peptide bonds that were previously published for the proteolysis of β-LG by trypsin (Table 1). These parameters are the enzyme selectivity [9] and the lag time [18]. In addition, the hydrolysis rate constants *k^j^* for the same peptide bonds were calculated using Equation (10) and are also presented in Table 1.

The values of lag time were used to assign peptide bonds to the one-stage or two-stage type of demasking, as described earlier [18]. The peptide bonds 8, 14, 40, 75, 138, 141, and 148 were assigned to the one-stage demasking, and the peptide bonds 83, 91, 124, and 135 to the two-stage demasking [18].

According to the values of enzyme selectivity and hydrolysis rate constants *k^j^*, the peptide bonds 8, 69/70, 75, 100/101, 138, 141, and 148 were referred to as the group of the most rapidly hydrolyzed bonds. The peptide bonds 20 and 60 were assigned to the group of the most slowly hydrolyzed bonds, and therefore, in our approach, it was assumed that they were not hydrolyzed at all. Thus, considering all these parameters, the trimeric regions in the β-LG polypeptide chain were selected as follows: 9–14, 15–40, and 41–69/70 (one-stage demasking); 76–83, 84–91, and 92–100/101 (two-stage demasking); 101/102–124, 125–135, and 136–138 (two-stage demasking) (Table 1).

### 2.4. Simulation of Peptide Release during β-LG Proteolysis by Trypsin

Here is an example of the dependence of the peptide concentrations on hydrolysis time for the intermediate fragments f(9–69/70) and f(9–40), as well as for the final products f(76–91) and f(101/102–124) (Figure 3a). The intermediate peptide products (ABC, AB, and BC) are first formed and then disappear due to hydrolysis of the internal enzyme-specific peptide bonds (Figure 3a). The final products (A, B, and C) only accumulate because they do not contain internal enzyme-specific peptide bonds. When the demasking step is a kinetically significant part of proteolysis, the concentrations of the proteolysis products may increase not immediately with the onset of proteolysis but with a lag phase [18]. This is also observed in the curves in Figure 3a, especially for the final peptide fragment f(101/102–124).

The data on the release of peptides during proteolysis are presented here on the degree of hydrolysis (Figure 3b). This way of presentation is more convenient for determining mechanisms by which various peptide bonds are demasked and hydrolyzed. The transformation from time to degree of hydrolysis practically does not change the concentration dependences for the intermediate products, but it does change the dependences for the final peptides. For them, the curves for fast-release peptides remain convex, while the curves for slow-release peptides become concave (peptide f(101/102–124) in Figure 3a,b).

For the intermediate products, we did not use any approximate functions for the concentration dependences but compared concentration curves based on the average degrees of hydrolysis *d_r_*. The degree of hydrolysis at which the main part of a given peptide is released was calculated for each of the intermediate products using Equation (8) (Table 2). The calculation methodology is described in detail in Section 4.3. 

For the final peptides, the curvature of the concentration curves was evaluated using Equation (9), which is a power function that allows us to determine the exponent *n* (Table 3). Thus, we compared the kinetic curves for the final products by simply comparing the parameter *n* for them.

Figure 4 and Figure 5 show examples of the calculated and experimental dependences of peptide concentrations on *d*.

The parameters *d_r_* for the intermediate peptides and *n* for the final peptides are shown in Table 2 and Table 3, respectively. To compare the simulation results with the experiment, the parameters *d_r_* and *n* were calculated from the experimental data [9] at published values of the degree of hydrolysis of 0, 1.5, 3, 4.5, 6, and 7.9%. For the same degrees of hydrolysis, the concentrations of peptide fragments were calculated using Equations (1)–(7).

The experimental and simulated concentration dependences of the intermediate trimeric peptides (ABC) differ from those of dimeric peptides (AB or BC) (Table 2, Figure 4). For all nine studied intermediate peptide fragments, trimeric peptides are released earlier than dimeric ones, and *d_r_* for trimeric peptide fragments is less than for dimeric ones. The difference in *d_r_* for the peptides of the same size, released with the participation of one-step and two-step demasking, is also different (Table 2, Figure 4). This difference in the release of ABC peptides (Figure 4a) is higher than that of AB peptides (Figure 4b). To determine the difference between the predicted *d_r_* and that determined from the experimental curves, we used seven peptides for which the experimental data were available (Table 2). The mean difference between the experimental and calculated values of *d_r_* was 0.6%, while the range of their variation was from 1.5 to 6.1%.

For the final peptides, the concentration curves were considered convex at *n* < 1, and at *n* > 1, the curves were considered concave. For the final peptides, it was found that for one group of peptides, the dependences were convex, and for the other, they were definitely concave (Table 3, Figure 5a). When peptide bonds are hydrolyzed by the two-stage demasking mechanism, the release of the final peptides gives concave curves. The convex curves correspond to the one-stage demasking. The *n* values for the simulated and experimental curves were compared with each other (Figure 5b).

For all three final peptides released with the one-stage demasking mechanism, lower *n* values were obtained compared to the other peptides released with the two-stage demasking. This was observed for both experimental and simulated *n*, although no assumptions about the presence of demasking were made when processing the experimental curves. The coefficient of proportionality between the calculated and experimental values of *n* was 1.25 ± 0.42 with the expected coefficient of 1 (Figure 5b). Thus, the agreement between the simulation and experiment was good.

For the proteolysis of β-LG by trypsin, the release of peptides was determined experimentally depending on the degree of hydrolysis [27]. In this publication, among the last released intermediate peptides were f(41–60), f(76–83), and f(125–138), as well as the peptides f(61–70 + 149–162) and f(41–70 + 142–162) bonded with the disulfide bond Cys_66_–Cys_160_ [27]. This is consistent with the fact that amino acid residues 76–138 in β-LG were noted as a trypsin-resistant core [28]. Implementation of demasking at the second stage of the two-stage demasking mechanism may be associated with the destruction of the peptide complex connected by a disulfide bridge and the degradation of the α-helical region of the polypeptide chain.

Approximately the same cleavage sites were identified by us in β-LG as peptide bonds cleaved by trypsin after two-stage demasking. The indices of such bonds were 20, 60, 83, 91, 124, and 135 [18] without considering the cleavage sites 69, 70, and 100, 101 with the -Lys-Lys- sequence. The peptide bonds were classified as hydrolyzable by the two-stage demasking mechanism if their hydrolysis occurred with significant time lags [18]. Thus, the evaluation of the lag phase on the kinetic curves gave the correct assignment of the bonds to the demasking mechanisms, which made it possible to make a fairly accurate prediction of the peptide release during proteolysis.

The experimental confirmation of the predicted patterns requires accurate measurements of the concentrations of released peptides. Among studies on this topic, we note the experiments in which peptide fractions [29,30] or individual peptides [19,27] were presented as functions of the degree of hydrolysis. In these studies, the change in proteolysis conditions was due to different concentrations of the enzyme and/or substrate [19,27,29,30], which strongly affect the hydrolysis time scale. The presentation of concentration dependences on the degree of hydrolysis made it possible to bring the dependences to the same scale.

The hydrolysis of casein by chymotrypsin at various *E*/*S* ratios with varying substrate concentrations was interpreted in the framework of the two-step proteolysis model [29]. It was taken into account that the change in the degree of hydrolysis may be the result of the hydrolysis of other peptides, which leads to a change in the course of the concentration dependence of the studied peptides. An illustrative example of such a change in concentration dependence from a convex function of time to a concave dependence on the degree of hydrolysis is presented in this work (Figure 3).

## 3. Discussion

The proposed method for predicting the release of peptides is a new one, based on the modeling of proteolysis, considering the gradual demasking of peptide bonds in the process of proteolysis. This approach was developed by us for the quantitative description of the proteolysis of various proteins by various proteases and was based mainly on kinetic data of the total hydrolysis of peptide bonds and tryptophan fluorescence [3,15,16,17,18,31]. The importance of taking into account demasking processes in the study of proteolysis was also shown by other analytical and physicochemical methods [20,32,33,34,35].

In the present work, it is shown that by knowing the rate constants of demasking and the rate constants of peptide bond hydrolysis, one can calculate the concentrations of peptide fragments. This is illustrated here by the example of the release of trimeric fragments, which are demasked by the mechanism of one-stage or two-stage demasking. The size of these fragments can be increased, and four-dimensional or longer fragments can be considered by obtaining the corresponding equations, similar to Equations (1)–(7).

An additional simplification in the calculations was that the hydrolysis of a small number of very slowly hydrolyzed bonds was not taken into account, and they were considered non-hydrolyzable (*k^i^* = 0). When processing the experimental data, the concentrations of the fragments formed during the hydrolysis of such bonds were low, and they were added to the concentrations of the initial peptides. Thus, the concentrations of dimeric peptides with internal bonds *i* = 20 and 60 slightly increased. In principle, it is possible to increase the size of peptide fragments and consider all specific bonds, including slowly hydrolyzed ones, which will increase the accuracy of peptide release prediction. However, the equations will be much more complicated.

The concentrations of peptide fragments were calculated using the example of β-LG proteolysis with trypsin as a practically important and experimentally well-studied case of proteolysis. For this case, the demasking parameters were determined using fluorescence spectroscopy [15,18]. The resulting peptides were identified, and their concentrations were determined at several values of the proteolysis time, which made it possible to build concentration dependences for the intermediate and final proteolysis products [9,27].

We proceeded from the fact that the concentration of the active enzyme is constant throughout the entire proteolysis, which allowed us to obtain analytical solutions for the kinetic schemes (Figure 1 and Figure 2). An alternative approach, taking into account changes in the concentration of the active enzyme during proteolysis, was considered earlier [31]. The effective Michaelis constant was expressed as a function of the degree of hydrolysis, and a new time variable was introduced, which made it possible to obtain an analytical solution as a function of this variable [31]. The change in the concentration of the active enzyme is associated with both the equilibrium inhibition of the enzyme by proteolysis products and the relatively slow irreversible inactivation of the enzyme during proteolysis [36]. The slow inactivation of the enzyme during proteolysis was shown by the example of the proteolysis of casein by chymotrypsin [36]. The slow inactivation of the enzyme explains the slowdown of proteolysis in the exponential model of proteolysis [37]. In the general case, numerical integration can be used, and the solution to the system of differential equations can be obtained by specifying a form of dependence of the active enzyme concentration on the proteolysis time.

The description of proteolysis using the concentrations of peptide bonds is based on a significant simplification, namely, on the assumption that the rate constants of hydrolysis of a given peptide bond are the same in various demasked peptides in which this bond is located. In the hydrolysis of the polypeptide chain by most enzymes, this assumption is justified since the binding sites of the active centers of these enzymes do not exceed the length of the hydrolyzed peptide fragments. An exception is the hydrolysis of peptides, in which specific bonds are located in the neighborhood. For the proteolysis of β-LG by trypsin, the effect of hydrolysis of neighboring specific bonds on each other can be primarily seen at the cleavage sites 69, 70, and 100, 101 with the amino acid sequence -Lys-Lys-. For a more accurate calculation in this case, it is better to use the description of proteolysis in terms of fragment concentrations, which are calculated using a computer program [5]. In this program, the influence of the peptide ends is considered an appropriate factor that reduces the rate constants in the short peptides. An alternative is to simplify the consideration of the -Lys-Lys- site as a single Lys-X peptide bond, as we did in the present study.

Identification and quantitative determination of peptides in hydrolysates can be effectively carried out by HPLC-MS methods [9,10,19]. The decrease in the concentration of a peptide bond during proteolysis is determined by summing the concentrations of all peptides formed as a result of the cleavage of this peptide bond. This method has been used to determine the enzyme selectivity for specific peptide bonds [10,19]. The proteolysis of whey proteins by the *Bacillus licheniformis* protease was analyzed by the hydrolysis of the individual bonds [19] and it was shown that more than half of the kinetic curves have a characteristic shape, indicating the presence of a demasking effect [20]. Fitting such curves with Equation (10) makes it possible to determine the hydrolysis rate constants *k^j^*. But these constants are only suitable for the hydrolysis of peptide bonds with one-stage demasking. Equation (13) from [18] could be used to determine the hydrolysis rate constants for the two-stage demasking. However, the application of this equation requires a more detailed measurement of the kinetic curves than was done in [9,10,19,20]. When selecting *k^j^* values for the two-stage demasking, we were not able to directly use the obtained *k^j^* values (Table 1), but they indicated which of the bonds hydrolyzed faster and which slower.

It is well known that changes in the conditions of proteolysis, such as temperature, pH, the presence of other molecules, etc., significantly change the total rate of the process. A study of the effect of pH on the kinetics of hydrolysis of the individual peptide bonds in whey proteins by the *Bacillus licheniformis* protease was carried out in [10], in which enzyme selectivity was the main kinetic parameter. Further studies are needed to answer how the physicochemical conditions of proteolysis affect the type of demasking, the rate constants of demasking, and the rate constants of hydrolysis of peptide bonds.

The results of predicting the release of peptides during the proteolysis of β-LG by trypsin show that the determination of the mechanism of demasking is of great importance. Determining the demasking mechanism for each peptide bond includes determining the number of demasking steps and the corresponding demasking rate constants. Apparently, the exact determination of the mechanism of demasking maybe even more important than the precise determination of the hydrolysis rate constants. This is favorable for simplifying proteolysis studies since it is not very difficult to determine the parameters of demasking by spectral methods [15,18]. Although the fluorescence spectroscopy method we used allows us to evaluate only the total effect of demasking for the entire polypeptide chain [15,18]. It would be interesting to know how the state of demasking of different regions of the polypeptide chain changes during proteolysis. The development of such an analytical method, in our opinion, will significantly advance the understanding of the details of proteolysis. 

The difference between prediction and experiment in modeling the release of peptides during proteolysis can be due to many reasons. Firstly, this is the imperfection of the model itself and additional simplifications made by us for the convenience of calculations. Secondly, this is insufficiently accurate knowledge of the values of the kinetic parameters included in the model. Third, the experimental data on peptide concentrations themselves contain errors. With all this, we state that a satisfactory agreement has been obtained between the simulation results and the experimental data. The modeling principles used are summarized in Table 4 to highlight the strengths and weaknesses of the calculation algorithm.

It should be noted that the methods for comparing the calculated and experimental data on the release of peptides during proteolysis have not been fully developed. In this work, for the intermediate peptides, it is proposed to compare the average degrees of hydrolysis at which these peptides are released. For the final peptides, the curvature of the concentration dependences on the degree of hydrolysis is compared. Further research should show if these methods are useful for studying peptide release or if others are needed.

The use of other enzyme-substrate pairs is necessary to more fully elucidate the role of peptide bond demasking in predicting the kinetics of peptide release. We hope that, taking into account the data for new enzyme–substrate pairs as well as for various proteolysis conditions (temperature, pH, enzyme, and substrate concentrations), the calculation algorithm (Table 4) will be improved in future works. 

## 4. Materials and Methods

### 4.1. Quantitative Modelling of Proteolysis with One-Stage Demasking

To calculate the relative concentrations *C*(*t*) of the peptide fragments (Figure 1) at different proteolysis times *t* [min], the following equations should be used: (1)$$[ABC]=\frac{k_{d}^{f}}{k^{i}+k^{j}-k_{d}^{f}}\left[e^{-k_{d}^{f}t}-e^{-\left(k^{i}+k^{j}\right)t}\right]$$
(2)$$[AB]=\frac{k_{d}^{f}k^{j}}{\left(k^{i}+k^{j}-k_{d}^{f}\right)\left(k^{i}-k_{d}^{f}\right)}e^{-k_{d}^{f}t}+\frac{k_{d}^{f}}{k^{i}+k^{j}-k_{d}^{f}}e^{-\left(k^{i}+k^{j}\right)t}-\frac{k_{d}^{f}}{k^{i}-k_{d}^{f}}e^{-k^{i}t}$$
(3)$$[BC]=\frac{k_{d}^{f}k^{i}}{\left(k^{i}+k^{j}-k_{d}^{f}\right)\left(k^{j}-k_{d}^{f}\right)}e^{-k_{d}^{f}t}+\frac{k_{d}^{f}}{k^{i}+k^{j}-k_{d}^{f}}e^{-\left(k^{i}+k^{j}\right)t}-\frac{k_{d}^{f}}{k^{j}-k_{d}^{f}}e^{-k^{j}t}$$
(4)$$[A]=1-\frac{k^{i}}{k^{i}-k_{d}^{f}}e^{-k_{d}^{f}t}+\frac{k_{d}^{f}}{k^{i}-k_{d}^{f}}e^{-k^{i}t}$$
(5)$$[B]=1-\left[1+\frac{k_{d}^{f}}{k^{i}+k^{j}-k_{d}^{f}}\left(1+\frac{k^{i}}{k^{j}-k_{d}^{f}}+\frac{k^{j}}{k^{i}-k_{d}^{f}}\right)\right]e^{-k_{d}^{f}t}-\frac{k_{1}}{k^{i}+k^{j}-k_{d}^{f}}e^{-\left(k^{i}+k^{j}\right)t}+\frac{k_{d}^{f}}{k^{j}-k_{d}^{f}}e^{-k^{j}t}+\frac{k_{d}^{f}}{k^{i}-k_{d}^{f}}e^{-k^{i}t}$$
(6)$$[C]=1-\frac{k^{j}}{k^{j}-k_{d}^{f}}e^{-k_{d}^{f}t}+\frac{k_{d}^{f}}{k^{j}-k_{d}^{f}}e^{-k^{j}t}$$

In the calculations, $k_{d}^{f}$ was 0.46 min^−1^ [18], and the hydrolysis rate constants were taken from Table 1 for the one-stage type of demasking. 

### 4.2. Quantitative Modelling of Proteolysis with Two-Stage Demasking

To calculate the relative concentrations of the peptide fragments (Figure 2), the following equation should be used:(7)$$C(t)=C_{0}+C_{1}e^{-k_{d}^{f}t}+C_{2}e^{-(k^{i}+k^{j})t}+C_{3}e^{-k_{d}t}+C_{4}e^{-k^{i}t}+C_{5}e^{-k^{j}t}$$
where the constant coefficients *C*_0_, *C*_1_, *C*_2_, and *C*_3_ are collected in Table 5 and the constant coefficients *C*_4_ and *C*_5_ in Table 6. $k_{d}^{f}$ was 0.46 min^−1^ [18], while *k_d_* was taken to be 0.15 min^−1^ in order to keep the ratio $k_{d}^{f}$/*k_d_* around 3 [18]. The hydrolysis rate constants were taken from Table 1 for the two-stage type of demasking.

### 4.3. Estimation of the Parameters for Concentration Dependences 

For the intermediate peptide fragments, the average hydrolysis degree of the peptide release *d_r_* was calculated using the following equation:(8)$$d_{r}=\sum_{i=1}^{6}d_{i}\times C(d_{i})/\sum_{i=1}^{6}C(d_{i}),$$
where *C*(*d_i_*) are the concentrations of peptide fragments determined at six hydrolysis degrees *d_i_* (0, 1.5, 3, 4.5, 6, and 7.9%). The same values of the degrees of hydrolysis were used in the calculations as in (9) with which the simulation results were compared. The experimental concentrations taken from Table S-4 [9] were divided by 50 to obtain the relative concentrations of the peptide fragments. 

For the final peptides, the parameter *n* was calculated using the following equation:(9)$$C(d_{i})=a{\left(d_{i}/7.9\right)}^{n},$$
where *a* is a constant factor and *n* is the exponent of the power function.

The concentration of hydrolysis products of the jth peptide bond is described by the following equation (Equation (12) from [18]):(10)$$N^{j}(t)=N_{0}\left[1-\frac{k^{j}e^{-k_{d}^{f}t}}{\left(k^{j}-k_{d}^{f}\right)}+\frac{k_{d}^{f}e^{-k^{j}t}}{\left(k^{j}-k_{d}^{f}\right)}\right],$$
where *N^j^* is the concentration of all peptides obtained as a result of hydrolysis of the jth peptide bond [18]. Equation (10) was used to determine the hydrolysis rate constant *k^j^* at a fixed value of $k_{d}^{f}$ = 0.46 min^−1^ from the experimental kinetic data [9]. 

## Figures and Tables

**Figure 1 ijms-24-11929-f001:**
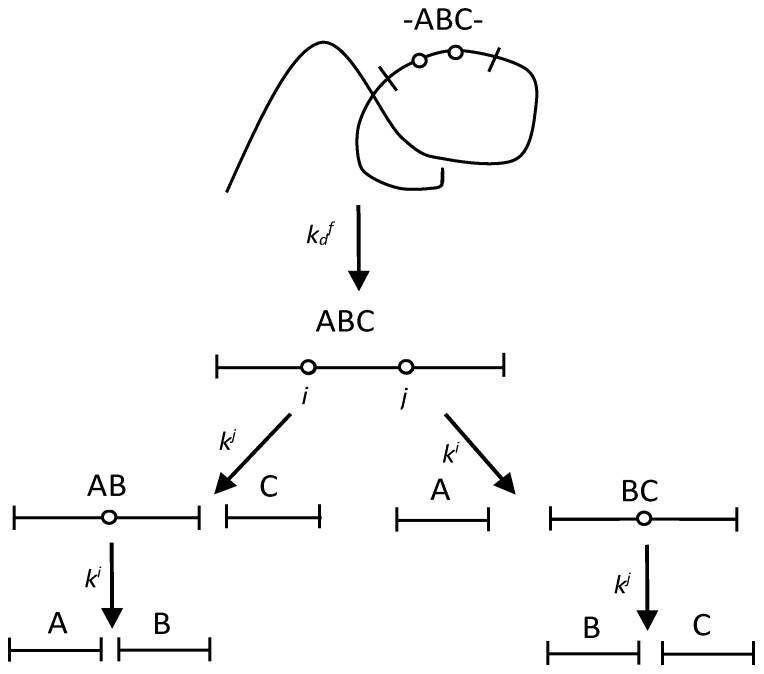
A scheme of peptide release for proteolysis with one-stage demasking.

**Figure 2 ijms-24-11929-f002:**
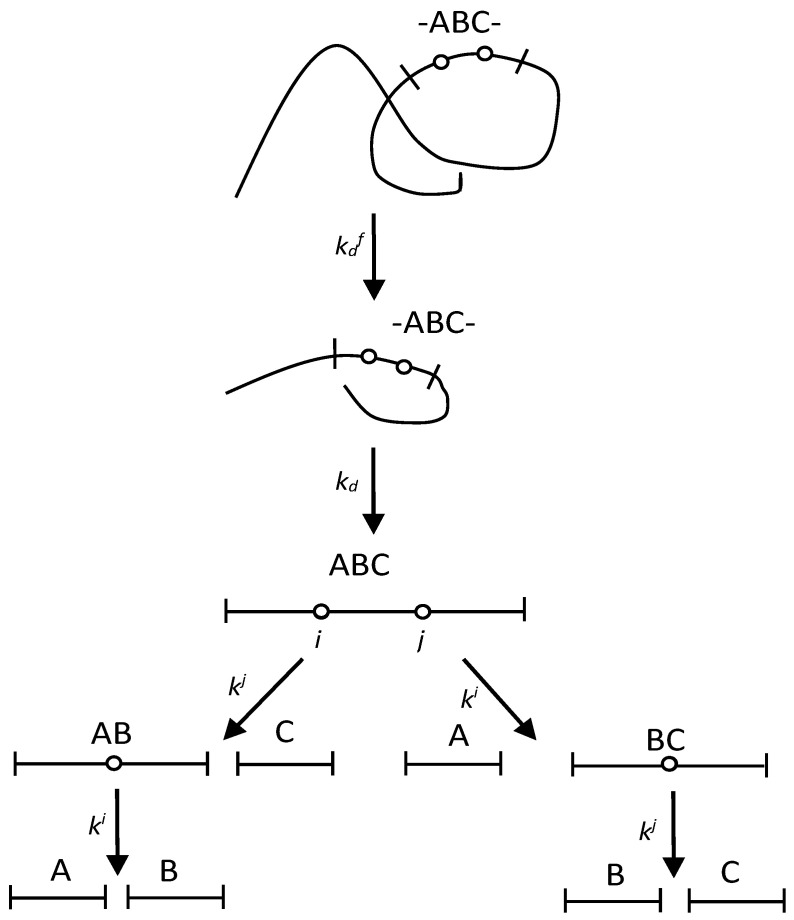
A scheme of peptide release for proteolysis with two-stage demasking.

**Figure 3 ijms-24-11929-f003:**
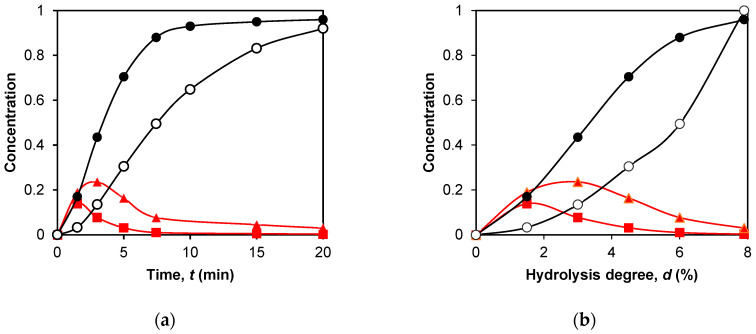
Simulation of peptide release during tryptic hydrolysis of β-LG: (**a**) calculated dependences of peptide concentrations on hydrolysis time for the intermediate peptides f(9–69/70) (■) and f(9–40) (▲), and for the final products of proteolysis f(76–91) (●) and f(101/102–124) (○); (**b**) calculated dependences of peptide concentrations on the degree of hydrolysis for the same peptide fragments.

**Figure 4 ijms-24-11929-f004:**
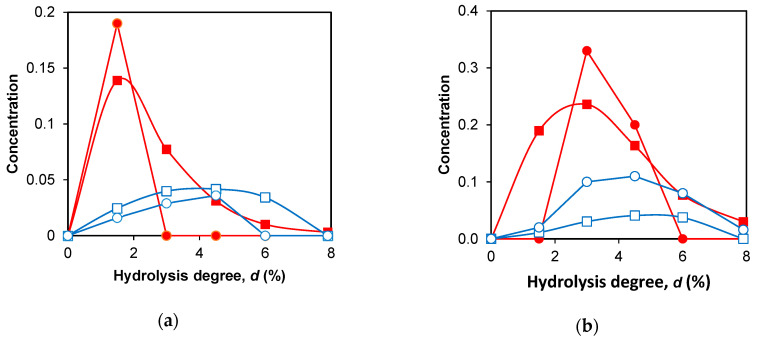
Simulation of peptide release during proteolysis: (**a**) calculated (■) and experimental (●) curves for peptide fragment f(9–79/70), ABC (one-stage demasking). Calculated (□) and experimental (○) curves for peptide fragment f(76–100/101), ABC (two-stage demasking); (**b**) calculated (■) and experimental (●) curves for peptide fragment f(9–40), AB (one-stage demasking). Calculated (□) and experimental (○) curves for peptide fragment f(76–91), AB (two-stage demasking).

**Figure 5 ijms-24-11929-f005:**
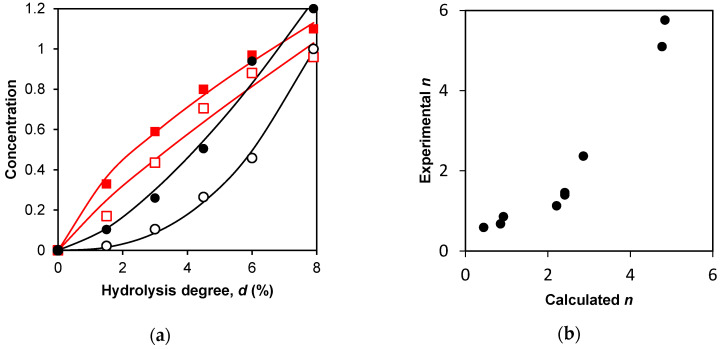
Processing of concentration curves for final products: (**a**) calculated (□) and experimental (■) concentration dependences for peptide f(9–14), A (one-stage demasking). Calculated (○) and experimental (●) concentration dependences for peptide f(76–83), A (two-stage demasking). Solid lines correspond to Equation (9); (**b**) correlation between experimental and calculated parameters *n* for 9 final products (Table 3).

**Table 1 ijms-24-11929-t001:** Kinetic parameters for tryptic hydrolysis of β-LG.

Bond Index *j*	Cleavage Site ^1^ $P_{1}P_{2}\;\frac{↓}{}\;P_{1}{}^{\prime }P_{2}{}^{\prime }$	Selectivity ^2^ (%)	$t_{lag}/t_{0}{\;}^{3,4}$	*k^j^/k* ^8 5,6^	Most Rapidly Hydrolyzed Bonds	Most Slowly Hydrolyzed Bonds	Peptide Fragments in Trimer Type of Demasking
8	MK-GL	13.7	0	>>1	+		9–14, 15–40, 41–69/70 One-stage demasking
14	QK-DL	7.4	0.33	1.2			
20	WY-SL		2.08	0.02		+	
40	LR-VY	9.9	0.15	3.1			
60	QK-WE	0.2	2.00	0.01		+	
69, 70		10.1 ^7^		21	+		
75	EK-TK	9.1	0.32	0.8	+		76–83, 84–91, 92–100/101 Two-stage demasking
83	FK-ID	2.9	1.21	0.4			
91	NK-VL	3.8	0.89	0.5			
100, 101		3.6 ^8^	0.85	1.1	+		
124	VR-TP	5.0	1.36	0.5			101/102–124, 125–135, 136–138 Two-stage demasking
135	EK-FD	1.6	1.94	0.05			
138	DK-AL	5.3	0.27	1.4	+		
141	LK-AL	9.4	0.18	2.3	+		
148	IR-LS	11.0	0.14	3.5	+		

^1^ Peptide bonds 69, 70 and 100, 101 with amino acid sequence -Lys-Lys- were considered as sites with single Lys bond and denoted as cleavage sites 69/70 and 100/101, respectively. ^2^ Values of enzyme selectivity were from [9]. ^3^ Values of lag time *t_lag_* were from [18]. ^4^ The characteristic time of hydrolysis *t*_0_ for the most rapidly hydrolyzed bond (*j* = 8) was 2.14 min. ^5^
*k^j^* values were calculated by Equation (10) at $k_{d}^{f}$ = 0.46 min^−1^. ^6^
*k*^8^ was 0.46 min^−1^. ^7^ Selectivity for *j* = 69 was presented. ^8^ Selectivity for *j* = 101 was presented.

**Table 2 ijms-24-11929-t002:** Simulation of the release of intermediate peptides.

Intermediate Peptide	Type of Demasking	Hydrolysis Rate Constants (min^−1^)	Calculated Values of *d_r_* (%) ^1^	Experimental Estimation of *d_r_* ^1^ (%)
f(9–69/70), ABC	One-stage ^2^	*k*^14^*=* 0.53, *k*^40^ *=* 1.41	2.6	1.5 ^3^
f(9–40), AB	One-stage		3.5	3.6
f(15–69/70), BC	One-stage		2.8	- ^4^
f(76–100/101) ABC	Two-stage ^5^	*k*^83^*=* 1, *k*^91^ *=* 1	3.9	3.4
f(76–91), AB	Two-stage		4.3	4.4
f(84–100/101), BC	Two-stage		4.3	4.7
f(101/102–138), ABC	Two-stage	*k*^124^*=* 2, *k*^138^ *=* 0.2	3.8	3.4
f(101/102–135), AB	Two-stage		4.1	- ^4^
f(125–138), BC	Two-stage		4.7	6.1

^1^ The average degree of hydrolysis was determined by Equation (8). ^2^ For the one-stage demasking, $k_{d}^{f}$ was 0.46 min^−1^ [18]. ^3^ This fragment was found only at one degree of hydrolysis *d* = 1.5% [9]. ^4^ These peptide fragments were not found [9]. ^5^ For the two-stage demasking, *k_d_* was 0.15 min^−1^ ($k_{d}^{f}$/*k_d_* = 3 [18]).

**Table 3 ijms-24-11929-t003:** Simulation of the release of final peptides.

Final Peptide	Type of Demasking	Hydrolysis Rate Constants (min^−1^)	Calculated Values of *n* ^1^	Experimental Estimation of *n* ^1^
f(9–14), A	One-stage ^2^	*k*^14^*=* 0.53, *k*^40^ *=* 1.41	0.85	0.68
f(15–40), B f(41–69/70), C	One-stage One-stage		0.92 0.44	0.86 0.59
f(76–83), A	Two-stage ^3^	*k*^83^*=* 1, *k*^91^ *=* 1	2.41	1.46
f(84–91), B	Two-stage		2.86	2.37
f(92–100/101), C	Two-stage		2.41	1.40
f(101/102–124), A	Two-stage	*k*^124^*=* 2, *k*^135^ *=* 0.2	2.21	1.13
f(125–135), B	Two-stage		4.84	5.76
f(136–138), C	Two-stage		4.77	5.10

^1^ The exponent of the power function *n* was determined using Equation (9). ^2^ For the one-stage demasking, $k_{d}^{f}$ was 0.46 min^−1^ [18]. ^3^ For the two-stage demasking, *k_d_* was 0.15 min^−1^ ($k_{d}^{f}$/*k_d_* = 3 [18]).

**Table 4 ijms-24-11929-t004:** A summary table showing the advantages and disadvantages of the used algorithm for predicting the release of peptides during proteolysis.

Modeling Principles	Advantages	Disadvantages
Proteolysis is described in terms of concentrations of peptide bonds	The technique for determining the concentrations of peptide bonds during proteolysis is well established	In the transition from the concentrations of peptides to the concentrations of peptide bonds, a part of the information is lost
Proteolysis is characterized by the set of hydrolysis rate constants for enzyme-specific bonds	Quantification of rate constants for hydrolysis of peptide bonds is well tested	The determination of the hydrolysis rate constants is limited by accuracy of determining experimental curves
Demasking of peptide bonds during proteolysis is considered	Accounting for demasking improves the accuracy of proteolysis description	Precise determination of the demasking rate constants requires the use of new analytical methods
The size of peptide fragments is limited by trimeric blocks	The equations describing the release of peptides are relatively simple	The model does not describe the release of four-dimensional or longer peptide blocks.
The hydrolysis of slowly hydrolyzed bonds is not considered	The volume of calculations is reduced	The release of minor peptide fragments is not predicted

**Table 5 ijms-24-11929-t005:** Coefficients *C*_0_, *C*_1_, *C*_2_, and *C*_3_ for the terms of Equation (7).

Peptide Fragment	Constant Term *C*_0_	$\mathrm{Coefficient}\;C_{1}\;\mathrm{at}\;e^{-k_{d}^{f}t}$	$\mathrm{Coefficient}\;C_{2}\;\mathrm{at}\;e^{-\left(k^{i}+k^{j}\right)t}$	$\mathrm{Coefficient}\;\mathrm{at}\;C_{3}\;e^{-k_{d}t}$
A	1	$\begin{array}{l} \frac{-k^{i}k_{d}}{\left(k_{d}-k_{d}^{f}\right)\left(k^{i}-k_{d}^{f}\right)} \end{array}$	0	$\frac{k_{d}^{f}k^{i}}{\left(k_{d}-k_{d}^{f}\right)\left(k^{i}-k_{d}^{f}\right)}$
B	1	$\frac{-k_{d}k^{i}k^{j}(k^{i}+k^{j}-2k_{d}^{f})}{\left(k_{d}-k_{d}^{f}\right)\left(k^{i}+k^{j}-k_{d}^{f}\right)\left(k^{j}-k_{d}^{f}\right)\left(k^{i}-k_{d}^{f}\right)}$	$\frac{k_{d}^{f}k_{d}}{\left(k^{i}+k^{j}-k_{d}^{f}\right)\left(k^{i}+k^{j}-k_{d}\right)}$	$\frac{k_{d}^{f}k^{i}k^{j}(k^{i}+k^{j}-2k_{d})}{\left(k_{d}-k_{d}^{f}\right)\left(k^{i}+k^{j}-k_{d}\right)\left(k^{j}-k_{d}\right)\left(k^{i}-k_{d}\right)}$
C	1	$\begin{array}{l} \frac{-k^{j}k_{d}}{\left(k_{d}-k_{d}^{f}\right)\left(k^{j}-k_{d}^{f}\right)} \end{array}$	0	$\frac{k_{d}^{f}k^{j}}{\left(k_{d}-k_{d}^{f}\right)\left(k^{j}-k_{d}^{f}\right)}$
AB	0	$\frac{k_{d}^{f}k_{d}k^{j}}{\left(k_{d}-k_{d}^{f}\right)\left(k^{i}+k^{j}-k_{d}^{f}\right)\left(k^{i}-k_{d}^{f}\right)}$	$\frac{k_{d}^{f}k_{d}}{\left(k^{i}+k^{j}-k_{d}^{f}\right)\left(k^{i}+k^{j}-k_{d}\right)}$	$\frac{k_{d}^{f}k_{d}k^{j}}{\left(k_{d}-k_{d}^{f}\right)\left(k^{i}+k^{j}-k_{d}\right)\left(k^{i}-k_{d}\right)}$
BC	0	$\frac{k_{d}^{f}k_{d}k^{i}}{\left(k_{d}-k_{d}^{f}\right)\left(k^{i}+k^{j}-k_{d}^{f}\right)\left(k^{j}-k_{d}^{f}\right)}$	$\frac{k_{d}^{f}k_{d}}{\left(k^{i}+k^{j}-k_{d}^{f}\right)\left(k^{i}+k^{j}-k_{d}\right)}$	$\frac{k_{d}^{f}k_{d}k^{i}}{\left(k_{d}-k_{d}^{f}\right)\left(k^{i}+k^{j}-k_{d}\right)\left(k^{j}-k_{d}\right)}$
ABC	0	$\frac{k_{d}^{f}k_{d}}{\left(k_{d}-k_{d}^{f}\right)\left(k^{i}+k^{j}-k_{d}^{f}\right)}$	$\frac{k_{d}^{f}k_{d}}{\left(k^{i}+k^{j}-k_{d}^{f}\right)\left(k^{i}+k^{j}-k_{d}\right)}$	$\frac{-k_{d}^{f}k_{d}}{\left(k_{d}-k_{d}^{f}\right)\left(k^{i}+k^{j}-k_{d}\right)}$

**Table 6 ijms-24-11929-t006:** Coefficients *C*_4_ and *C*_5_ for the terms of Equation (7).

Peptide Fragment	$\mathrm{Coefficient}\;C_{4}\;\mathrm{at}\;e^{-k^{i}t}$	$\mathrm{Coefficient}\;C_{5}\mathrm{at}\;e^{-k^{j}t}$
A	$\frac{-k_{d}^{f}k_{d}\left[\left(k^{i}+k^{j}\right)\left(k^{i}+k^{j}-k_{d}^{f}-k_{d}\right)+k_{d}^{f}k_{d}\right]}{\left(k^{i}-k_{d}^{f}\right)\left(k^{i}-k_{d}\right)\left(k^{i}+k^{j}-k_{d}^{f}\right)\left(k^{i}+k^{j}-k_{d}\right)}$	0
B	0	0
C	0	$\frac{-k_{d}^{f}k_{d}\left[\left(k^{i}+k^{j}\right)\left(k^{i}+k^{j}-k_{d}^{f}-k_{d}\right)+k_{d}^{f}k_{d}\right]}{\left(k^{j}-k_{d}^{f}\right)\left(k^{j}-k_{d}\right)\left(k^{i}+k^{j}-k_{d}^{f}\right)\left(k^{i}+k^{j}-k_{d}\right)}$
AB	$\frac{-k_{d}^{f}k_{d}\left[\left(k^{i}+k^{j}\right)\left(k^{i}+k^{j}-k_{d}^{f}-k_{d}\right)+k_{d}^{f}k_{d}\right]}{\left(k^{i}-k_{d}^{f}\right)\left(k^{i}-k_{d}\right)\left(k^{i}+k^{j}-k_{d}^{f}\right)\left(k^{i}+k^{j}-k_{d}\right)}$	0
BC	0	$\frac{-k_{d}^{f}k_{d}\left[\left(k^{i}+k^{j}\right)\left(k^{i}+k^{j}-k_{d}^{f}-k_{d}\right)+k_{d}^{f}k_{d}\right]}{\left(k^{j}-k_{d}^{f}\right)\left(k^{j}-k_{d}\right)\left(k^{i}+k^{j}-k_{d}^{f}\right)\left(k^{i}+k^{j}-k_{d}\right)}$
ABC	$\frac{-k_{d}^{f}k_{d}\left[\left(k^{i}+k^{j}\right)\left(k^{i}+k^{j}-k_{d}^{f}-k_{d}\right)+k_{d}^{f}k_{d}\right]}{\left(k^{i}-k_{d}^{f}\right)\left(k^{i}-k_{d}\right)\left(k^{i}+k^{j}-k_{d}^{f}\right)\left(k^{i}+k^{j}-k_{d}\right)}$	$\frac{-k_{d}^{f}k_{d}\left[\left(k^{i}+k^{j}\right)\left(k^{i}+k^{j}-k_{d}^{f}-k_{d}\right)+k_{d}^{f}k_{d}\right]}{\left(k^{j}-k_{d}^{f}\right)\left(k^{j}-k_{d}\right)\left(k^{i}+k^{j}-k_{d}^{f}\right)\left(k^{i}+k^{j}-k_{d}\right)}$

## Data Availability

The data presented in this study are available on request from the corresponding author.

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
