# Peer review of "Modeling of the Peptide Release during Proteolysis of β-Lactoglobulin by Trypsin with Consideration of Peptide Bond Demasking"

_ijms, 2023, doi:10.3390/ijms241511929_

Round 1
Reviewer 1 Report
This manuscript entitled "Modeling of the peptide release during proteolysis of beta-lactoglobulin by trypsin" reports novel and interesting data. The author has published previously, along with other authors, a series of relative articles, which are cited in this manuscript. I read the cited articles, and it seems to me that there is a systematic and continuous approach to the main idea of the present work i.e., the introduction of a specific algorithm, which may be found helpful in the biotechnology of manufacturing antiallergic foodies.
The algorithm presented herein contributes significantly to calculate the concentrations of time depended hydrolysis, in peptide fragments of a chosen protein (beta- lactoglobulin), by based on model di- and/or tri- peptide fragments, as well as to detect the mechanism of demasking of peptide bonds before their hydrolysis. The corresponding rate constants can be estimated also through the proposed algorithm. Additionally, author succeeded to achieve an almost acceptable agreement between experimental data and simulation results.
The methods and the results are well described; the discussion analyzes and comments properly the results. In my opinion, discussion should be re-written and include also future perspectives of this work.
Another suggestion to the author is to unify Figures 1 and 2 in one by incorporating the reaction pathway to -ABC-.
Therefore, I can suggest the publication of the manuscript after a minor revision, by considering the abovementioned comments, as well as after improving the text by performing an extensive editing of English language.
Suggestions for an extensive editing of English language of the text.
Author Response
I would like to thank the reviewer for a very careful consideration of the manuscript.
Reviewer 1: In my opinion, discussion should be re-written and include also future perspectives of this work.
Reply: The Discussion section has been rewritten and, in particular, the following paragraph on the future perspectives has been added:
The use of other enzyme-substrate pairs is necessary to more fully elucidate the role of peptide bond demasking in predicting the kinetics of peptide release. We hope that taking into account the data for new enzyme–substrate pairs, as well as for various proteolysis conditions (temperature, pH, enzyme and substrate concentrations), the calculation algorithm (Table 4) will be improved in future works.
Reviewer 1: Another suggestion to the author is to unify Figures 1 and 2 in one by incorporating the reaction pathway to -ABC-.
Reply: Unfortunately, we were unable to combine the two figures into one, since the overall drawing is too illegible and inconvenient for readers.
Reviewer 1: I can suggest the publication of the manuscript after a minor revision, by considering the abovementioned comments, as well as after improving the text by performing an extensive editing of English language.
Reply: Extensive editing of the English language was carried out.
Reviewer 2 Report
In the manuscript entitled “Modeling of the peptide release during proteolysis of β-lacto- 2 globulin by trypsin” the authors show and underline the potential of using proteolysis models in predicting peptide fragmentation patterns. By considering the demasking process and employing a two-step proteolysis model, the authors achieved good agreement between their predictions and experimental observations. In general, the manuscript is well written and concise. However, before it can be accepted for publication, some points must be clarified, and authors need to address the following main issues:
1. The authors should provide a section/period in the discussion/conclusions where they explain the scientific experimental impact of the study.
2. The title should be more focused on the effective message of the study.
3. The impact of certain factors, such as pH, temperature and the presence of co-factors/co-substrates on the proteolysis process should be taken into account. Have the authors considered the effect of these factors could potentially have on the kinetics of peptide release and the accuracy of model predictions? In my opinion this part should be address in discussion.
4. To enhance the readers' comprehension of the methodology, I suggest the inclusion of a table that illustrates the advantages and disadvantages of the proposed method. Such a table would serve as a valuable reference, providing a concise overview of the strengths and limitations of the approach.
Minor revisions
1) It would be appropriate to improve the English form and pay attention to punctuation.
2) Improve the resolution of figure1 and figure 2.
In the manuscript entitled “Modeling of the peptide release during proteolysis of β-lacto- 2 globulin by trypsin” the authors show and underline the potential of using proteolysis models in predicting peptide fragmentation patterns. By considering the demasking process and employing a two-step proteolysis model, the authors achieved good agreement between their predictions and experimental observations. In general, the manuscript is well written and concise. However, before it can be accepted for publication, some points must be clarified, and authors need to address the following main issues:
1. The authors should provide a section/period in the discussion/conclusions where they explain the scientific experimental impact of the study.
2. The title should be more focused on the effective message of the study.
3. The impact of certain factors, such as pH, temperature and the presence of co-factors/co-substrates on the proteolysis process should be taken into account. Have the authors considered the effect of these factors could potentially have on the kinetics of peptide release and the accuracy of model predictions? In my opinion this part should be address in discussion.
4. To enhance the readers' comprehension of the methodology, I suggest the inclusion of a table that illustrates the advantages and disadvantages of the proposed method. Such a table would serve as a valuable reference, providing a concise overview of the strengths and limitations of the approach.
Minor revisions
1) It would be appropriate to improve the English form and pay attention to punctuation.
2) Improve the resolution of figure1 and figure 2.
Author Response
I would like to thank the reviewer for a very careful consideration of the manuscript.
Main revisions:
1. The scientific and experimental impact of the studyis formulated in the following paragraph:
The results of predicting the release of peptides during the proteolysis of b-LG by trypsin show that the determination of the mechanism of demasking is of great importance.Determining the demasking mechanism for each peptide bond includes determining the number of demasking steps and the corresponding demasking rate constants.Apparently, the exact determination of the mechanism of demasking may be even more important than the precise determination of the hydrolysis rate constants.This is favorable for the simplifying proteolysis studies, since it is not very difficult to determine the parameters of demasking by spectral methods [15,18]. Although the fluorescence spectroscopy method we used allows us to evaluate only the total effect of demasking for the entire polypeptide chain [15,18].It would be interesting to know how the state of demasking of different regions of the polypeptide chain changes during proteolysis.The development of such an analytical method, in our opinion, will significantly advance the understanding of the details of proteolysis.
2. The title of the article is now: Modeling of the peptide release during proteolysis of b-lactoglobulin by trypsin with consideration of peptide bond demasking.
3. The following paragraph was added to the discussion:
It is well known that changes in the conditions of proteolysis, such as temperature, pH, the presence of other molecules, etc., significantly change the total rate of the process. A study of the effect of pH on the kinetics of hydrolysis of the individual peptide bonds in whey proteins by the Bacillus licheniformis protease was carried out in [10], in which enzyme selectivity was the main kinetic parameter. Further studies are needed to answer how the physicochemical conditions of proteolysis affect the type of demasking, the rate constants of demasking, and the rate constants of hydrolysis of peptide bonds.
4. The new table was included in the discussion: Table 4. A summary table showing the advantages and disadvantages of the used algorithm for predicting the release of peptides during proteolysis.
Minor revisions:
1) Extensive editing of the English language was carried out.
2) The resolution of figure1 and figure 2 was increased.
Round 2
Reviewer 2 Report
the authors satisfied my issues